:ᐰ: PLOS | ONE

# Prediction of poor outcome after hypoxic-ischemic brain injury by diffusion-weighted imaging: A systematic review and meta-analysis

Ruili Wei[1⚬], Chaonan Wang[2⚬], Fangping He[1], Lirong Hong[3], Jie Zhang[4], Wangxiao Bao[1], Fangxia Meng[1], Benyan Luo[1]*

1 Department of Neurology, Brain Medical Centre, First Affiliated Hospital, Zhejiang University School of Medicine, Hangzhou, China, 2 Department of Geriatrics, Shulan (Hangzhou) Hospital, Hangzhou, China, 3 Department of Rehabilitation Medicine, Zhejiang Provincial People's Hospital, People's Hospital of Hangzhou Medical College, Hangzhou, China, 4 Department of Rehabilitation, Hangzhou Hospital of Zhejiang CAPR, Hangzhou, China

⚬ These authors contributed equally to this work.
* luobenyan@zju.edu.cn

**Data Availability Statement:** All relevant data are within the manuscript and its Supporting Information files.

## Abstract

Accurate prediction of the neurological outcome following hypoxic–ischemic brain injury (HIBI) remains difficult. Diffusion-weighted imaging (DWI) can detect acute and subacute brain abnormalities following global cerebral hypoxia. Therefore, DWI can be used to predict the outcomes of HIBI. To this end, we searched the PubMed, EMBASE, and Cochrane Library databases for studies that examine the diagnostic accuracy of DWI in predicting HIBI outcomes in adult patients between January1995 and September 2019. Next, we conducted a comprehensive meta-analysis using the Meta-DiSc and several complementary techniques. Following the application of inclusion and exclusion criteria, a total of 28 studies were included with 98 data subsets. The overall sensitivity and specificity, with 95% confidence interval, were 0.613(0.599–0.628) and 0.958(0.947–0.967), respectively, and the area under the curve was 0.9090. Significant heterogeneity among the included studies and a threshold effect were observed (p<0.001). Different positive indices were the major sources for the heterogeneity, followed by the anatomical region examined, both of which significantly affected the prognostic accuracy. In conclusion, we demonstrated that DWI can be an instrumental modality in predicting the outcome of HIBI with good prognostic accuracy. However, the lack of clear and generally accepted positive indices limits its clinical application. Therefore, using more reliable positive indices and combining DWI with other clinical predictors may improve the diagnostic accuracy of HIBI.

## Introduction

Hypoxic–ischemic brain injury (HIBI) occurs secondary to multiple events that cause hypoxia or hypoperfusion like cardiac arrest, respiratory failure, hanging, drowning or severe

**Funding:** This work was financially supported by the Administration of traditional Chinese medicine project of Zhejiang Province (2018ZA067) and National Natural Science Foundation of China (81671143).

**Competing interests:** The authors have declared that no competing interests exist.

hypotension as a result of oxygen and nutrient deprivation [1]. Despite recent heath care advances, HIBI remains one of the principle causes of death and long-term disability world-wide. Specifically, the toll of the neurological recovery, possible complications and rehabilitation imposes a huge socioeconomic burden on individuals as well as the health care system as a whole [2, 3]. Therefore, identifying patients who can likely achieve a favorable or poor neurological outcome will significantly impact the patient prognosis and facilitate informed health care decisions.

Diminished brain-stem or extensor reflex, day three motor response, and day one cortical somatosensory evoked potentials (SSEPs), as well as serum neuron specific enolase (NSE) during the first three days and early myoclonic status epilepticus were used to predict poor HIBI outcome [4–6]. However, the emerging use of therapeutic hypothermia for the management of comatose cardiac arrest patients has decreased the utility of the above mentioned markers [7–10]. Particularly, therapeutic hypothermia involves the use of sedatives and neuromuscular blockers during the induction and normothermia phases which render the prognostic predictors less reliable, especially those based on clinical examination [7, 11]. Therefore, developing a more accurate assessment of early-stage HIBI patients is urgently needed.

Neuroimaging approaches like magnetic resonance imaging (MRI), Diffusion-weighted imaging (DWI), and computed tomography (CT) are commonly used diagnostic techniques for exploring brain structure and function [12]. Nevertheless, CT and conventional MRI frequently underestimate the degree of brain injury in acute HIBI [13, 14]. On the other hand, DWI provides a more accurate diagnostic alternative in acute or subacute HIBI [15] and enables precise estimation of disease degree by calculating the apparent diffusion coefficient (ADC) [16, 17]. Moreover, DWI has been proven valuable in therapeutic hypothermia or sedated patients [18, 19]. Previous studies have investigated the diagnostic and prognostic value of DWI in HIBI; however, the sensitivity and specificity of DWI as a clinical tool were inconsistent among the different studies [14, 20–22]. In this study, we performed a meta-analysis of previously published literature to re-evaluate the diagnostic value of DWI in predicting HIBI outcomes.

## Methods

### Study design

In this study, we performed a comprehensive literature research in PubMed, EMBASE, and the Cochrane Library databases for DWI from January 1995 to September 2019. We examined the diagnostic value of DWI in predicting HIBI outcomes using the following keywords: ("diffusion-weighted magnetic resonance images" or "diffusion magnetic resonance" or "DW-MRI" or "DW magnetic resonance images" or "diffusion-weighted imaging" or "diffusion MRI" or diffusion-weighted MRI") and ("anoxia" or "ischemia" or "hypoxia" or "heart arrest" or "cardiac arrest" or "postoperative complication" or "respiratory insufficiency" or "resuscitation" or "drowning") and ("prognosis" or "outcome"). In addition, we also examined the reference section of all examined articles for additional reports. In some cases, we had to contact the corresponding authors to seek the original data sets if the necessary information could not be extracted online. From each study, we gathered and analyzed the following information: patients' baseline demographic characteristics (gender, age, hypothermia treatment and outcome assessment), study design (prospective or retrospective), experimental protocol, elapsed interval between HIBI and brain MRI, DWI imaging protocol (magnetic field strength, b-value, and positive indices), and the diagnostic results (i.e. the true-positive, false-positive, false-negative, and true-negative results).

In order to investigate the predictive power of DWI on HIBI outcome, we only analyzed studies that examined the neurological outcome in terms of the five cerebral performance categories (CPCs) or an equivalent [5, 23]. A CPC score of 1 indicated full recovery; 2 indicated moderate disability, 3 indicated severe neurological disability with preserved consciousness, 4 indicated comatose or vegetative state patients, 5 indicated death. Next, the outcome was classified into poor and good according to the CPC scores (3–4 or 4–5 versus1-2 or 1–3, respectively).

## Inclusion and exclusion criteria

The inclusion criteria included English language clinical prognostic DWI articles that were published in indexed journals and studies investigating adult HIBI patients (>/ = 14 years). Also, we included studies reporting various causes of HIBI, provided that each condition resulted in the common endpoint of generalized cerebral hypoxia or global hypoperfusion. Finally, only studies with complete data sets (i.e.the number of true/false negatives and positives for poor outcome prediction) were included. This was essential to enable the calculation of outcome variables with confidence intervals (CIs). Exclusion criteria included published abstracts, case reports, review articles and studies involving 10 patients or less, as well as patients with HIBI secondary to stroke, trauma, intracranial infection, sepsis, and/or metabolic dysfunction.

We confirmed the quality of the included studies using the Quality Assessment of Diagnostic Accuracy Studies (QUADAS-2) tool as detailed previously [24]. Two primary investigators were responsible for data collection and quality assessment in an independent manner.

## Statistical analysis

A Chi-square test and the inconsistency index ($I^2$) were used to estimate the heterogeneity between enrolled studies. A $P < 0.1$ or $I^2 > 50\%$ indicated the presence of heterogeneity[25]. If heterogeneity was recorded, a binary regression model with random coefficients was used to determine the diagnostic performance [26]. The summary receiver operating characteristic (SROC) curve, and area under the curve (AUC) were used to predict the outcome of HIBI [27].

The threshold effect was determined from the "shoulder-arm" shape of the ROC curve [28]. A correlation between the logit of sensitivity and the logit of (1—specificity) was computed by the Spearman correlation coefficient to assess the existence of a threshold effect, and a $P < 0.05$ indicated a positive threshold effect [29]. Next, we performed a meta-regression analysis and subgroup analysis to investigate factors that could possibly lead to heterogeneity and explored their possible impact on diagnostic accuracy [30].

The analysis of heterogeneity test, the threshold effect and the diagnostic performance, as well as meta-regression and subgroup analyses were all carried out by Meta-DiSc (version 1.4) [31]. On the other hand, publication bias was assessed by an asymmetry test and Deeks' funnel plot using Stata (version 12.0). An inverted symmetrical funnel plot with $P > 0.05$ indicated the lack of publication bias [32].

## Results

From January 1995 to September 2019, we collected a total of 4042 records from the different data bases. After applying the inclusion and exclusion criteria, a total of 28 studies were included in this meta-analysis (Fig 1).

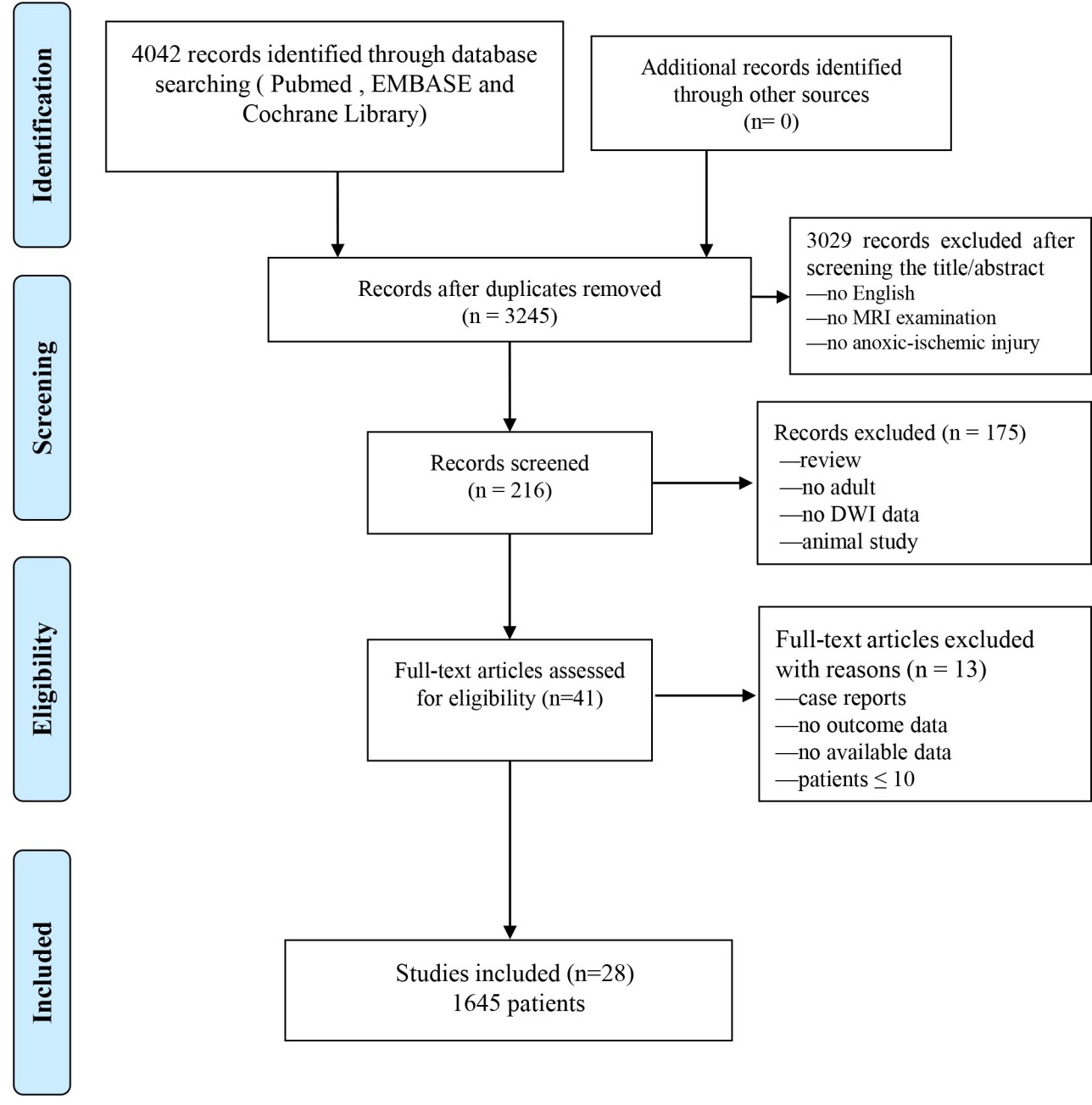

**Fig 1. Flow chart representing the scheme of our study design.**

## Study features and quality assessment

The clinical features and baseline characters of patients in each study examined are presented in Table 1. A total of 1,645 patients (age range between 14 and 89 years) were enrolled from the 28 studies. The average number of patients in each included study was 59 (range 14–172). Among the investigated studies, 14 were conducted prospectively and the remaining 14 studies were retrospective. Five studies collected data from out-of-hospital cardiac arrest (OHCA) patients, while the other 23 studies included OHCA and in-hospital cardiac arrest (IHCA) patients. Hypothermia treatment was administered to all patients in 11 studies (n = 600), and

**Table 1. Study designs and baseline patient characteristics.**

| Author, year, reference | Type | IHCA or OHCA | No. of patients | Males, % | Mean age, years [±SD] or median (IQR range) | Treatment with hypothermia | Definition of poor outcome | Timing of outcome assessment |
|---|---|---|---|---|---|---|---|---|
| Barrett,2007[33] | Retro | Mix | 18 | 10(56%) | 62 (49~73) | no | CPC3-5 | Death or hospital discharge |
| Bevers,2018[34] | Retro | Mix | 78 | 49(63%) | 53 ± 17 | yes | CPC4,5 | hospital discharge |
| Choi,2010 [35] | Pro | OHCA | 39 | 28 (71.8%) | 49.1(18~89) | 15/39 | CPC3-5 | 3 months |
| Choi,2018 [36] | Pro | Mix | 14 | 10 (71.4%) | 43.4 ± 15.6 | 8/14 | CPC 3–5 | at discharge. |
| Cronberg,2011 [37] | Pro | Mix | 22 | N/A | N/A | yes | CPC4-5 | 6 months |
| Els,2004 [14] | Pro | Mix | 12 | N/A | 53 (27~71) | no | CPC4-5 | 6 months |
| Greer,2012 [38] | Retro | Mix | 80 | 49(61%) | 57±16 | 14/80 | mRS5 | 3 months |
| Greer,2013 [39] | Pro | Mix | 80 | 49(61%) | 62 (IQR 46–70) | 14/80 | mRS4-5 | 6 months |
| Hirsch,2015 [40] | Pro | Mix | 68 | 44 (64.7%) | 56 ± 15 | 37/68 | CPC4-5 | 6 months |
| Hirsch,2016[22] | Retro | Mix | 125 | 82(66%) | 58 ± 16 | 77/125 | CPC4-5 | Day 14 or at discharge |
| Jarnum,2009 [18] | Pro | Mix | 20 | 11(55%) | 57.8(14~81) | yes | CPC3-5 | 6 months |
| Jeon,2017 [41] | Retro | Mix | 39 | 27 (69%) | 52.2±16.5 | yes | CPC3-5 | 6 months |
| Kim,2012[42] | Retro | OHCA | 43 | 29 (67.4%) | 57±17.6 | yes | CPC3-5 | 6 months |
| Kim,2013[43] | Retro | OHCA | 51 | 38 (74.5) | 63(IQR, 42–72) | 45/51 | CPC3-5 | 6 months |
| Kim,2016[44] | Retro | OHCA | 110 | 83 (75.5%) | 59 (47–70) | 100/110 | CPC3-5 | 6 months |
| Luyt,2012[45] | Pro | Mix | 57 | 40 (70%) | 52 ± 18 | 36 /57 | GOS-E1-4 | 12 months |
| Mettenburg,2016[46] | Retro | Mix | 33 | N/A | 54(24–80) | yes | CPC4,5 | at discharge |
| Mlynash,2010[47] | Pro | Mix | 32 | 23(72%) | 55.5±17.3 | 21/32 | CPC4-5 | 6 months |
| Moon,2018[48] | Pro | Mix | 96 | 66 (68%) | 52±16 | yes | CPC3-5 | 6 months |
| Oren,2019[49] | Retro | Mix | 38 | 20 (52.6%) | 52.8(18–87) | N/A | CPC4-5 | 6 months |
| Park,2015[50] | Pro | Mix | 19 | 16 (84.2%) | 54.6±18.7 | yes | CPC3-5 | at discharge |
| Reynolds,2017[51] | Retro | Mix | 69 | 37(54%) | 60 (IQR 50, 73) | 60/69 | CPC3-5 | at discharge |
| Ryoo,2015[52] | Retro | OHCA | 172 | 117 (68.0%) | 54.7 ± 16.0 | yes | CPC3-5 | at discharge |
| Topcuoglu,2009[53] | Retro | Mix | 22 | 14(61%) | 56±16.9 | no | CPC4-5 | 6 months |
| Velly,2018[54] | Pro | Mix | 150 | 97 (65%) | 51 ±16 | 110/150 | CPC3-5 | 6 months |
| Wallin,2018[55] | Pro | Mix | 46 | 31 (67%) | 68 (IQR 59–76) | yes | CPC3-5 | 6 months |
| Wijman,2009[19] | Pro | Mix | 32 | N/A | N/A | yes | CPC4-5 | 6-month |
| Wu,2009[21] | Retro | Mix | 80 | 49(61%) | 57±16 | 14/80 | mRS4-5 | 6 months |

**Note**: N/A = data unavailable; Retro = retrospective study; Pro = prospective study; OHCA = out-of-hospital cardiac arrest; IHCA = in-hospital cardiac arrest; Mix = OHCA or IHCA; CPC = cerebral performance categories; GOS-E = expand the Glasgow outcome scale score; mRS = Modified Rankin Scale. IQR = interquartile range.

only some patients in the other 13 studies (n = 551). In the remaining 4 studies, hypothermia treatment was not offered or not mentioned to patients (n = 90). Outcome was assessed at hospital discharge, death or within several weeks in 8 studies; at 3 months in 2 studies, and at 6 months or more in 18 studies. A poor outcome was defined as CPC 3–5 in 14 studies or as CPC 4–5 in another 10 studies, and according to other scoring systems in the remaining 4 studies.

**Table 2. Characteristics of the imaging protocol of the enrolled studies.**

| Study | Elapsed interval | Field strength | b-value (s/mm²) | Positive Index | Sensitivity | Specificity |
|---|---|---|---|---|---|---|
| **Barrett,2007** | 72 h (IQR,22–229) | 1.5T | 1000s | DWI abnormalities | 0.700 | 0.750 |
| **Bevers,2018** | 4 (IQR 3–5) | N/A | N/A | whole brain ADC signal intensity | 0.191 | 1.000 |
| | | | | 15% total brain volume with ADC signal intensity < 650 mm²/s | 0.362 | 1.000 |
| **Choi,2010** | 52.9h ± 37.5 | 1.5T | 0 /1000s | mixed pattern of brain injury | 0.769 | 0.923 |
| | | | | mean ADC value of frontal cortex | 0.714 | 1.000 |
| | | | | parietal cortex | 0.857 | 1.000 |
| | | | | temporal cortex | 0.643 | 1.000 |
| | | | | occipital cortex | 0.929 | 1.000 |
| | | | | precentral cortex | 0.857 | 1.000 |
| | | | | postcentral cortex | 0.714 | 1.000 |
| | | | | caudate nucleus | 0.643 | 1.000 |
| | | | | putamen | 0.929 | 1.000 |
| | | | | thalamus | 0.857 | 1.000 |
| **Choi,2018** | 3h | 1.5 T | 0/1000s | HSI on early DWI | 0.909 | 1.000 |
| **Cronberg,2011** | 106h(IQR93-118) | 1.5T or 3T | 0 /1000s | extensive brain injury | 0.579 | 1.000 |
| **Els,2004** | 16 h (4–32) | 1.5T | N/A | multiple cortical areas abnormalities | 1.000 | 1.000 |
| **Greer,2012** | 48 h (IQR 0–10h) | 1.5T | 0 /1000s | any imaging abnormality | 0.985 | 0.462 |
| | | | | basal ganglia abnormalities | 0.791 | 0.692 |
| | | | | cortical abnormalities | 0.955 | 0.462 |
| | | | | cerebellar abnormalities | 0.612 | 0.538 |
| **Greer,2013** | 48 h (IQR 0–10h) | 1.5T | 0 /1000s | bilateral hippocampal hyperintensities | 0.273 | 1.000 |
| **Heradstveit,2011** | 3h | 1.5T | 0 /1000s | DWI abnormalities | 0.000 | 1.000 |
| | 32h | | | DWI abnormalities | 1.000 | 1.000 |
| | 96h | | | DWI abnormalities | 1.000 | 1.000 |
| **Hirsch,2015** | 77h (IQR58-144h) | 1.5T | 0/ 1000s | qualitative MRI scoring system | 0.600 | 1.000 |
| | | | | DWI score (25~192h) | 0.725 | 1.000 |
| **Hirsch,2016** | 69 h± 25 | 1.5T | 0 /1000s | >10% Brain volume with ADC<650x10⁻⁶ mm²/s | 0.717 | 0.909 |
| | | | | >22% Brain volume with ADC<650x10⁻⁶ mm²/s | 0.522 | 1.000 |
| **Jarnum,2009** | 123 h (39–251h) | 1.5T or 3T | 0 /1000s | diffuse signal abnormalities | 0.824 | 1.000 |
| **Jeon,2017** | 175(117.5–240)min | 1.5 T | 1000s | positive high signal on DW-MRI | 0.813 | 1.000 |
| **Kim,2012** | 45.8h(IQR,36.8–52.4) | 3.0T | 1000s | ADC value of frontal cortex | 0.625 | 1.000 |
| | | | | parietal cortex | 0.656 | 1.000 |
| | | | | temporal cortex | 0.563 | 1.000 |
| | | | | occipital cortex | 0.906 | 1.000 |
| | | | | precentral cortex | 0.656 | 1.000 |
| | | | | postcentral cortex | 0.719 | 1.000 |
| | | | | caudate nucleus | 0.469 | 1.000 |
| | | | | putamen | 0.781 | 1.000 |
| | | | | thalamus | 0.625 | 1.000 |
| | | | | cerebellum | 0.563 | 1.000 |
| | | | | pons | 0.469 | 1.000 |
| **Kim,2013** | 46 h (IQR,37–52) | 3.0T | 1000s | MCS of frontal region | 0.700 | 1.000 |
| | | | | occipital region | 0.900 | 1.000 |
| | | | | parietal region | 0.825 | 1.000 |
| | | | | rolandic region | 0.800 | 1.000 |

*(Continued)*

**Table 2.** (Continued)

| Study | Elapsed interval | Field strength | b-value (s/mm²) | Positive Index | Sensitivity | Specificity |
|---|---|---|---|---|---|---|
| | | | | temporal region | 0.625 | 1.000 |
| | | | | BG region | 0.750 | 1.000 |
| | | | | LMEAN of frontal region | 0.650 | 1.000 |
| | | | | occipital region | 0.625 | 1.000 |
| | | | | parietal region | 0.625 | 1.000 |
| | | | | rolandic region | 0.725 | 1.000 |
| | | | | temporal region | 0.550 | 1.000 |
| | | | | BG region | 0.500 | 1.000 |
| | | | | LMIN of frontal region | 0.725 | 1.000 |
| | | | | occipital region | 0.750 | 1.000 |
| | | | | parietal region | 0.825 | 1.000 |
| | | | | rolandic region | 0.675 | 1.000 |
| | | | | temporal region | 0.625 | 1.000 |
| | | | | BG region | 0.425 | 1.000 |
| Kim,2016 | 53 h(46–72) | 1.5T or 3T | N/A | mean ADC of the entire brain | 0.506 | 1.000 |
| | | | | median ADC of the entire brain | 0.494 | 1.000 |
| | | | | LADCV | 0.747 | 1.000 |
| | | | | DC-LADCV | 0.892 | 1.000 |
| Luyt,2012 | 11 d(7–17) | 1.5 or 3.0T | N/A | mean diffusivity values in nine grey regions | 0.837 | 0.875 |
| Mettenburg,2016 | 4d | 1.5T | 1000s | diffuse pattern of restricted diffusion (diffuse brain injury) | 0.238 | 0.917 |
| | | | | diffuse pattern of gyral edema | 0.429 | 0.917 |
| | | | | restricted diffusion in basal ganglia (any) | 0.667 | 0.917 |
| | | | | restricted diffusion in the hippocampi | 0.286 | 1.000 |
| Mlynash,2010 | 80 h(IQR, 55–117) | 1.5T | 0 /1000s | extensive cortical lesion pattern | 0.800 | 1.000 |
| | | | | abnormalities in basal ganglia | 0.867 | 0.500 |
| | | | | abnormalities in brainstem | 0.200 | 1.000 |
| Moon,2018 | 17±14h | 3.0T | 1000s | PV500 > 6.25% | 0.720 | 1.000 |
| | 17±14h | | | PV400 >2.50% | 0.640 | 1.000 |
| | 17±14h | | | Mean ADC< = 726× $10^{-6}$ mm²/s | 0.440 | 1.000 |
| | 77±23h | | | PV400>1.66% | 0.792 | 1.000 |
| | 77±23h | | | Mean ADC< = 627× $10^{-6}$ mm²/s | 0.208 | 1.000 |
| Oren,2019 | 2.9d (1~5d) | 1.5 or 3.0T | 0 /1000s | abnormalities on DWI/ADC | 0.815 | 0.545 |
| Park,2015 | 2h (1.5–3.3h) | 1.5T | 0/1000s | overall qualitative DWI scores | 1.000 | 1.000 |
| | | | | DWI scores of Cortex | 0.917 | 1.000 |
| | | | | DWI scores of Cortex + DGN | 1.000 | 1.000 |
| Reynolds,2017 | 4d (IQR3-6) | 1.5 or 3.0T | 1000s | ≥2.8% diffusion restriction of the entire brain at an ADC of ≤650 × $10^{-6}$ mm²/s | 0.682 | 1.000 |
| | | | | ADC changes in the thalamus at an ADC threshold of ≤650 × $10^{-6}$ mm²/s | 0.183 | 1.000 |
| Ryoo,2015 | 2.0d [1.0–3.0] | 1.5 or 3.0T | 1000s | positive DWI finding or regional brain injury of frontal cortex | 0.729 | 0.963 |
| | | | | parietal | 0.814 | 0.963 |
| | | | | temporal | 0.686 | 0.981 |
| | | | | occipital | 0.771 | 0.963 |
| | | | | basal ganglia or thalamus | 0.466 | 1.000 |
| | | | | cerebellum | 0.314 | 1.000 |
| | | | | brain stem | 0.025 | 1.000 |
| | | | | MRI positive finding | 0.864 | 0.926 |

*(Continued)*

**Table 2.** (Continued)

| Study | Elapsed interval | Field strength | b-value (s/mm$^2$) | Positive Index | Sensitivity | Specificity |
|---|---|---|---|---|---|---|
| Topcuoglu,2009 | 136.8h±108 | 1.5T | 1000s | extensive cortical lesion pattern | 0.875 | 1.000 |
| Velly,2018 | 13d(7-18d) | 1.5 T or 3T | 0 /1000s | FLAIR-DWI overall score | 0.402 | 1.000 |
| | | | | FLAIR-DWI cortex score | 0.333 | 1.000 |
| | | | | FLAIR-DWI cortex plus deep grey nuclei score | 0.368 | 1.000 |
| Wallin,2018 | 4 d(IQR,4–5) | 1.5 T or 3T | 0 /1000s | acute hypoxic-ischemic lesions | 0.773 | 0.625 |
| Wijdicks,2001 | 144h (24~360) | 1.5T | 0 /1000s | diffuse signal abnormalities | 1.000 | 1.000 |
| Wijman,2009 | 49–108h | 1.5T | 0 /1000s | >10% brain volume with ADC<650x10$^{-6}$ mm$^2$/s | 0.810 | 1.000 |
| Wu,2009 | 2d (IQR 0–10d) | 1.5T | 0 / 1000s | whole-brain median ADC | 0.409 | 1.000 |

**Note:** DC-LADCV = the relative volume of the dominant (biggest) cluster of the low-ADC voxels. HIS = high signal intensity; LADCV = the relative volume of voxels with ADC values less than the predefined ADC threshold; LMEAN = lowest mean ADC; LMIN = lowest minimum ADC; MCS = Maximum cluster size; PV = % voxels with ADC values below the predefined ADC thresholds.

MRI parameters of each study are presented in Table 2. Briefly, a 1.5-T MRI scanner was used in 15 studies, a 3.0-T MRI scanner was used in 3 studies and both scanners were used in another 9 studies (Table 2). In the final study, the type of scanner was unclear. With respect to b-values in the DWI, a single b-value of 1000 s/mm$^2$ was used in 9 studies; b-values of both 0 s/mm$^2$ and 1000 s/mm$^2$ were used in 15 studies. The b-value(s) used were unclear in the remaining 4 studies. The mean elapsed interval between MRI and HIBI ranged from 2 hours to 13 days (Table 2). Further, among the 28 studies, 15 studies used qualitive MRI-positive indices for their analysis, 10 studies used a quantitative index, 1 study used both and the final 3 studies used a semi-quantitative index [40, 50, 54]. Within the same study, multiple sets of data were considered as different DWI-positive indices. Therefore, we had 98 data subsets for meta-analysis (Table 2).

In 21 studies, MRI analyses were performed in a blinded manner. In one study, the investigators were not blind to the examined groups and it was not indicated in the remaining 6 studies. For ethical reasons, the image analysts were blinded to clinical information and outcome in 21 studies, but the clinical treatment team was blinded to the imaging analysis results in only 3 studies [22, 45, 54]. Further, it is worth mentioning that all of the examined studies had a relatively small study population which may affect the reliability of the results obtained in the current work. Therefore, we performed a quality assessment test using the QUADAS-2 tool (Fig 2, S1 Fig). Among the 28 studies, 13 studies demonstrated patient selection bias risk and applicability concerns. With regards to the index test, a total of 8 studies had bias risk; while 8 studies had applicability concerns (Fig 2, S1 Fig).

## Diagnostic performance

The overall sensitivity and specificity were 0.613 (95% CI, 0.599–0.628) and 0.958 (95% CI, 0.947–0.967), respectively (Fig 3A and 3B). In the SROC analysis, the AUC and Q-index were 0.9090 and 0.8410, respectively, thereby, indicating a good diagnostic accuracy (Fig 3C). For individual studies, the sensitivity ranged from 2.5% to 100%, and their specificity ranged from 46% to 100%. These results indicate a significant heterogeneity among the examined studies (Fig 3A and 3B).

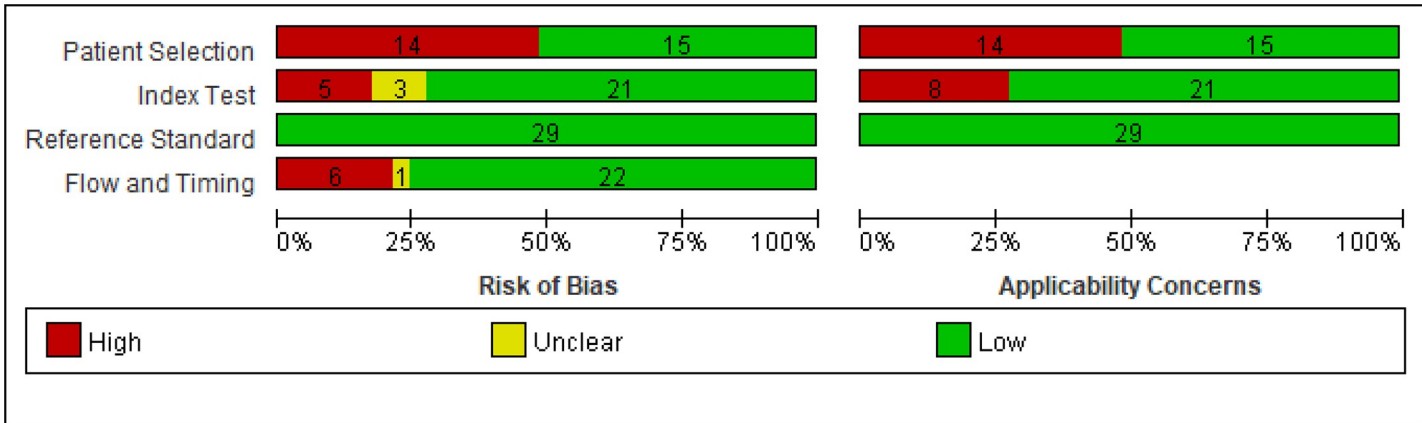

**Fig 2. Evaluation of the included studies using Quality Assessment of Diagnostic Accuracy Studies (QUADAS-2) tool.** Bias risk and applicability concerns were analyzed in all studies and categorized into high (red), low (green) and unclear (yellow).

## Assessment of study heterogeneity

A significant heterogeneity was detected in the sensitivities and specificities of the included studies (*P < 0.001*). The ROC curve demonstrated a "shoulder-arm" shape indicative of a threshold effect (Fig 3D). Additional analysis revealed a significant linear correlation between

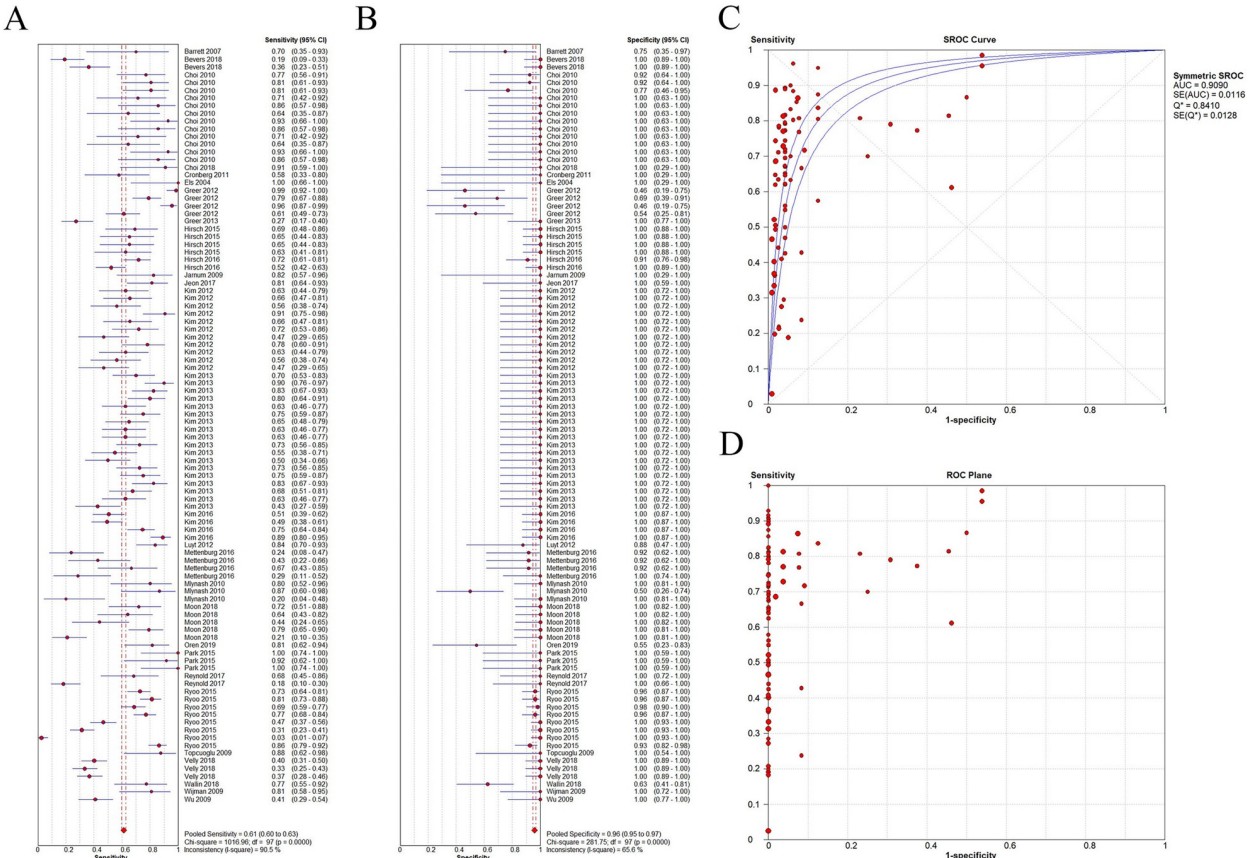

**Fig 3. Diagnostic performance of the included studies.** A,B: Forest plot demonstrating the sensitivity and specificity of individual studies arranged in alphabetical order. C Summary receiver operating characteristic (SROC) curve, D Receiver operating characteristic (ROC) plane respectively. CI: confidence intervals.

the logit of sensitivity and the logit of (1—specificity) (r = 0.539, P < 0.001), thereby, confirming a threshold effect which resulted in the notable heterogeneity. This led us to hypothesize that different positive indices (cutoff values) were the major source of heterogeneity.

Next, we explored other factors that can cause heterogeneity via meta-regression analysis using the following predictor variables: study type (prospective or retrospective), patient category (OHCA or IHCA), hypothermia treatment (present/absent), poor outcome definition, timeframe for outcome assessment, study bias (blinding), elapsed time until brain MRI, field strength, b-value, test index (qualitative or quantitative), and the examined brain region. The patient category, test index and the examined brain region were selected by multivariate meta-regression analysis as significant predictor variables that can affect heterogeneity.

## Subgroup analysis

Next, we carried out a subgroup analysis on the different study subsets (Table 3). Among the examined brain regions, the cortical region had the highest diagnostic accuracy, followed by the basal ganglia region (moderate diagnostic accuracy). While, the other brain regions (cerebellum, brain stem, and hippocampus) showed low diagnostic accuracy (P = 0.0049). Interestingly, the diagnostic accuracy was similar when scanning the cortical regions only and the whole brain.

The pooled data revealed that qualitative and quantitative analysis methods had a similar diagnostic accuracy, while the semi-quantitative analysis had lower diagnostic accuracy (P = 0.0018) [40, 50, 54]. Interestingly, the elimination of Velly et al.[54], in which the DWI examination was performed 6 days after onset (7–18 day), from semi-quantitative group can

**Table 3. Subgroup analysis among the different study subsets.**

| Study characteristics | No of subsets | Pooled sensitivity(95% CI) | Pooled specificity(95% CI) | P |
|---|---|---|---|---|
| **Total** | 98 | 0.613(0.599–0.628) | 0.958(0.947–0.967) | |
| **Region measured** | | | | 0.0049 |
| **Global** | 33 | 0.611(0.587–0.636) | 0.951(0.93–0.966) | |
| **Cortex** | 41 | 0.712(0.689–0.733) | 0.973(0.958–0.984) | |
| **Basal Ganglia** | 16 | 0.582(0.541–0.623) | 0.928(0.888–0.958) | |
| **Others** | 8 | 0.301(0.259–0.344) | 0.968(0.931–0.988) | |
| **Index test** | | | | 0.0018 (0.0009) |
| **qualitive** | 32 | 0.635(0.612–0.659) | 0.904(0.880–0.925) | |
| **quantitive** | 56 | 0.628(0.608–0.648) | 0.995(0.987–0.999) | |
| **Semi-quantitive(all)** | 10 | 0.472(0.427–0.518) | 1.000(0.984–1.000) | |
| **Semi-quantitive(<7d)** | 7 | 0.739(0.658–0.810) | 1.000(0.973–1.000) | |
| **Time of MRI examination** | | | | 0.1426 |
| **~1d** | 9 | 0.767(0.694–0.829) | 1.000(0.960–1.000) | |
| **2~6d** | 85 | 0.627(0.611–0.642) | 0.953(0.941–0.963) | |
| **>6d** | 4 | 0.425(0.376–0.475) | 0.991(0.949–1.000) | |
| **OHCA or IHCA** | | | | 0.0001 |
| **OHCA** | 53 | 0.646(0.627–0.665) | 0.984(0.973–0.991) | |
| **IHCA or MIX** | 45 | 0.566(0.543–0.589) | 0.925(0.903–0.942) | |
| **Co-index** | | | | 0.0008 |
| **DWI** | 98 | 0.613(0.599–0.628) | 0.958(0.947–0.967) | |
| **co-index** | 6 | 0.862(0.823–0.895) | 1.000(0.977–1.000) | |

**Note:** OHCA = out-of-hospital cardiac arrest; Mix = OHCA or IHCA (in-hospital cardiac arrest)

change the significance of the results. Specifically, the diagnostic accuracy of semi-quantitative group would have been significantly better than that of the qualitive or quantitative index within the first 7 days after HIBI (p = 0.0008).

Further, the MRI examination time is also an important factor affecting the diagnostic accuracy except for test index. The analysis of different time points demonstrated that the diagnostic accuracy of DWI within 6 days of onset was higher than that of after 6 days, but it did not reach statistical significances (p = 0.1426). Moreover, the imaging diagnostic accuracy was higher in the OHCA patients than the IHCA or mixed (OHCA/IHCA) patients (P = 0.0001).

Furthermore, we observed that DWI imaging indices combined with other predictors (co-index) like brain CT[41], EEG[34], motor response[34, 40] or other MRI modalities [54] produced significantly improved diagnostic accuracy (Table 3; P = 0.0008).

## Publication bias

There was no evidence of publication bias (P = 0.19) as revealed by the symmetric distribution of diagnostic odds ratio against (effective sample size)$^{-1/2}$(S2 Fig).

## Discussion

In this study, we analyzed the efficiency of DWI in predicting a poor outcome of HIBI. Our meta-analysis results demonstrated that DWI is an accurate imaging tool for predicting HIBI outcome with high specificity (95.9%). On the other hand, individual studies showed significant heterogeneity in terms of sensitivity and specificity. This heterogeneity was primarily attributed to the threshold effect, in addition to the test index, the region imaged and the patients' categorization (OHCA or IHCA). The different imaging protocols, signal characteristics and anatomic regions measured accounted for different positive indices which affected the overall imaging diagnostic accuracy.

Our meta-analysis results indicated that the diagnostic accuracy varied substantially according to the region being assessed. The cortical region demonstrated the highest diagnostic accuracy, followed by the basal ganglia with moderate accuracy. Therefore, DWI signal abnormalities or ADC reduction can be significantly influenced by the anatomical region examined. Several studies showed that DWI signal abnormalities or ADC reduction were also time dependent [21, 35, 47]. In poor-outcome patients, Mlynash et al. confirmed that cortical structures exhibited the most profound ADC reductions, which were observed as early as 1–2 days after the HIBI and reached a nadir 3–5 days after the HIBI. Therefore, Wijman et al proposed that the ideal prognostic window is between 49 and 108 hours after HIBI [19]. Our subgroup analysis also showed that diagnostic accuracy during the 6 days that follows HIBI was higher than that after the 6 days period. Interestingly, the diagnostic accuracy during the first 24 hours after HIBI was not less stringent than other studies acquiring the results of DWI during the ideal prognostic window. This could be attributed to the use of sensitive indicators, like abnormal high signal presence on DW-MRI during the acute window rather than extensive abnormality.

Since MRI signal is affected by the region and time of detection, it is particularly important to select an appropriate diagnostic strategy and index. Typically, post-ischemic MRI images display cortical or basal ganglia hyperintensity in DWI sequences [56, 57]. Following HIBI, the presence of large or extensive multilobar alterations on DWI MRI images has been correlated with poor outcome [58]. The apparent diffusion coefficient (ADC) value has been widely used to quantitatively assess the progression of ischemia when using DWI. In HIBI, several ADC methods, such as determining the whole-brain ADC value, quantifying the region with low ADC, or calculating the lowest ADC value in a specific brain area have been previously used to

predict patient outcomes [21, 43, 47]. However, we observed that different research centers used different predictors. Therefore, there was a lack of clear and generally accepted positive indices, especially for the quantitative indices. Consequently, it was difficult for those indicators to be widely applied among the different research centers. In agreement, the 2015 guidelines of the European Society of Intensive Care Medicine and European Resuscitation Council also highlighted the limitations of studies using MRI to prognosticate following HIBI; noting the lack of homogeneity in radiological definitions of imaging findings [11]. Thus, there is a need for a specific DWI index that can be used concisely in the clinic.

The semi-quantitative method (qualitative MRI scoring system) has been successfully developed as a tool to predict the outcome following perinatal asphyxia, and has been reported to provide an accurate index for HIBI severity following postanoxic coma [40, 59]. Our study showed that qualitative brain MRI scoring system was also good for predicting the outcome of the HIBI and may be an ideal DWI index for clinical use. Future well-designed, large-scale studies should be carried out to confirm the best positive index. The combination of qualitative and quantitative methods, or machine-based auto-analysis can be potential directions for future studies.

Although DWI could predict the outcome of HIBI with good prognostic accuracy, there are still several limitations of solely depending on it. Guidelines from professional societies advocated neuroimaging was recommended only in combination with other predictors [11]. Our pooled data also showed that DWI examination combined with other predictors could improve diagnostic accuracy [34, 40, 41, 54]. Therefore, the integration of DWI data with other prognostic markers such as serial neurological assessments, physiological tests, serum marker levels or other model MRI examination in the future could be instrumental for the prediction of HIBI outcome. This model will ultimately affect the patients' care strategies.

In conclusion, in this study we performed a systemic review and meta-analysis to assess the ability of DWI in predicting poor outcome in HIBI. Our results indicated that DWI can accurately predict the poor outcome of HIBI. Nevertheless, this meta-analysis had various limitations. First, patients with implanted devices like pacemakers or implantable cardioverter defibrillators (ICDs), or other metallic objects could not undergo the conventional MRI. For example, only 21/514 (4.1%) cardiopulmonary arrest survivors underwent subsequent brain MRIs, which may reflect a patient selection bias [33]. However, this issue was resolved in more recent studies [19, 40].

Second, our meta-analysis included different populations, like OHCA or IHCA patients, or patients who did or did not undergo hypothermia treatment (and the reporting of outcome assessment and the timeframe thereof), and reflects a wide variability in case characteristics. In addition, the strategies for active treatment withdrawal differed between studies. These differences can partly be responsible for the heterogeneity of our results.

Third, the retrospective nature of 14 from the 28 included studies, the relatively small sample size in each individual study and the absence of proper blinding measures in almost all studies (25/28) could have led to studies with a low quality of evidence. Further, the exclusion of non-English articles could have limited the strength of our meta-analysis. Therefore, future research should include more studies to confirm our results and evaluate the predictive value of DWI in global brain anoxia.

## Conclusion

Our meta-analysis demonstrated that DWI can accurately predict the outcome of HIBI. However, the diagnostic accuracy is influenced by the region measured and time of MRI acquisition. Furthermore, the lack of clear and generally accepted positive indexes limits its clinical

application. The use of a more reliable positive index and combining DWI with other predictors may help to improve the accuracy of diagnosis.

## Supporting information

**S1 PRISMA Checklist. PRISMA checklist.**
(DOC)

**S1 Fig. Risk of bias and applicability concerns summary.** Review authors' judgements about each domain for each included study.
(TIF)

**S2 Fig. DEEKS funnel analysis to assess the publication bias.**
(TIFF)

## Author Contributions

**Conceptualization:** Ruili Wei, Benyan Luo.

**Data curation:** Chaonan Wang, Fangping He, Lirong Hong, Wangxiao Bao.

**Formal analysis:** Lirong Hong, Jie Zhang.

**Investigation:** Chaonan Wang, Wangxiao Bao.

**Methodology:** Ruili Wei, Jie Zhang, Fangxia Meng.

**Project administration:** Chaonan Wang, Fangping He.

**Resources:** Lirong Hong, Wangxiao Bao.

**Software:** Jie Zhang, Fangxia Meng.

**Supervision:** Benyan Luo.

**Validation:** Fangxia Meng.

**Writing – original draft:** Ruili Wei.

**Writing – review & editing:** Ruili Wei, Benyan Luo.

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
