## [Decision Letter · Decision Letter 0]

2 Sep 2019

PONE-D-19-17572

Prediction of neurological outcome after hypoxic-ischemic brain injury by diffusion-weighted imaging: A systematic review and meta-analysis

PLOS ONE

Dear Dr. Luo,

Thank you for submitting your manuscript to PLOS ONE. After careful consideration, we feel that it has merit but does not fully meet PLOS ONE’s publication criteria as it currently stands. Therefore, we invite you to submit a revised version of the manuscript that addresses the points raised during the review process.

We would appreciate receiving your revised manuscript by Oct 17 2019 11:59PM. To enhance the reproducibility of your results, we recommend that if applicable you deposit your laboratory protocols in protocols.io, where a protocol can be assigned its own identifier (DOI) such that it can be cited independently in the future. For instructions see: http://journals.plos.org/plosone/s/submission-guidelines#loc-laboratory-protocols

We look forward to receiving your revised manuscript.

Kind regards,

Chiara Lazzeri

Academic Editor

PLOS ONE

Journal Requirements:

Reviewers' comments:

Reviewer's Responses to Questions

**Comments to the Author**

1. Is the manuscript technically sound, and do the data support the conclusions?

Reviewer #1: Partly

Reviewer #2: Partly

2. Has the statistical analysis been performed appropriately and rigorously? 

Reviewer #1: No

Reviewer #2: I Don't Know

3. Have the authors made all data underlying the findings in their manuscript fully available?

Reviewer #1: Yes

Reviewer #2: Yes

4. Is the manuscript presented in an intelligible fashion and written in standard English?

Reviewer #1: No

Reviewer #2: Yes

5. Review Comments to the Author

Reviewer #1: The authors investigated an interesting topic that can have relevant clinical meaning. The methodology is accurate and rigorous. But I have concerns regarding many points that I will detail below.

The first point is that I am very puzzled in the statistical analysis, because if each author has chosen different cut-offs to get 100% specificity hat is FPR = 0 it cannot be said that the specificity of the method is this and not very variable. The analysis shows in fact that there is no shared index and that each author has decided his own, therefore the negativity of other cofactors in conditioning the specificity and a false deduction The conclusion of the work that can be deduced is that each work used different indices, whereas I would prefer to see cumulative analysis of homogeneous set of data. For example It is well known that MRI findings changes truough the time, so diffent index can be usefult at different time, so the specificity is not of the MRI but of index and Is time dependent.

More in details

Title: according to the results please add in the title “poor”

Abstracts: “explicit” please find a synonymous.

Please use prognostic instead of diagnostic

Introduction:

pg 3:

“disease modifying agents” this definition is not appropriate for HIE,

“normothermia phase which render the prognostic predictors less reliable[7].” This is not true, please have a loo to recent Bibliography, such as Scarpino et al., 2018, Resuscitation

pg 4: developing a more accurate assessment of acute-stage HIBI patients is urgently needed.: I do not agree with these, because performing MRI is not possible usually in the first 24 hours

Nevertheless, CT and conventional MRI frequently underestimate the degree of brain injury in acute HIBI [12: being a systematic review I think not appropriate referring to a reference of 1999.

Methods

Pg 4:

In this study, we performed a comprehensive literature research in PubMed, EMBASE, and the Cochrane Library databases for DWI from January 1995 to December 2018.: this point seems to be very illogical for two mean reasons 1) TTM have been introduced about 2002 2) DWI is not a technique introduced so early. Actually only one reference is dated 2001.

Pg 5: The outcome was classified into poor and good according to the CPC scores (3-4 or 4-5 versus1-2 or 1-3, respectively). Alternatively, we classified the outcomes into CPC 4–5 versus 1–3, if the CPC thresholds were not defined.

Please clarify this point, the authors extrapolate the data from the table presented by authors according to the cut off they used. I do not agree to attribute a priori a cut off if this is not specified and the studies not reporting clear indication should be not considered in the analysis.

Table 2: I’m very surprised to find in the table studies in which idex test has specificity and sensitivity of 100%. He authors reviewed the quality of the study and how can include these studies in their computation?

DISCUSSION

PG 17

“Given this finding, the semi-quantitative method was used by only 2 studies”

Please rephrase this sentence, this is a systematic review, not an original paper reporting data about a sample of patients.

PG 18

“Further, during brain damage, Wu et al., demonstrated an initial ADC reduction in the striatum and thalamus, followed by the cortex and the subcortical white matter. This DWI pattern could be an indication of ongoing tissue damage due to secondary apoptotic processes, and thus various brain structures can respond differently to ischemic injury [22].”

This sentence is not useful, in this kind of paper the authors should report consideration about cumulative data, is not a narrative report.

Pg 18-19

“However, our pooled analysis did not demonstrate that sensitivity and specificity of DWI were time dependent. The meta-regression analysis showed that the elapsed time between HIBI and brain MRI examination was not a factor in heterogeneityInterestingly, their diagnostic accuracy was not less stringent than other studies acquiring the DWI data during the ideal time window. This suggests that the time of the MRI is not an important factor in determining diagnostic accuracy.

This is an example of what I have underlined in the first part of my comments; this results is true if we evaluated the value of specificity, in this case 100%, but this finding would be right only if in all the paper the authors had used the same parameters to reach the best predictive power at every window. In fact the the studies reported by the authors [20, 38, 44, 48, 51, 52] all used different measure od MRI , so the message that time dependence of MRI is not a factor is not true. MRI is a time dependent test and for every time frame require different measure to reach the best predicitve power.

PG 19

“Therefore, DWI examination could be carried out in a wider timeframe than other prognostic strategies like clinical examination, myoclonus and status myoclonus, electroencephalogram, or biomarkers”

Again this is not completely true, EEG and SEP can be performed in any time windows, and offer also the advantage to be more available in every clinical setting, can be recorded bed-side and repeated more time.

CONCLUSION

They need to be completely rephrased according to the revised version of the manuscript according to the point raised.

Reviewer #2: The authors gave a review and meta-analysis of different diffusion weighted imaging studies predicting neurological recovery in HIBI patients. They find that DWI has a high diagnostic accuracy, but clinical application is limited due to the high variety in study design of the analyses articles.

Although diagnostic properties for HIBI patients are of great interest to clinicians, the manuscript in its current format has, in my opinion, limited added value for clinical practice.

In my opinion, the authors have performed many statistical tests, but with limited adjustments to the clinical, technical and pathofysiological background of the DWI analyses. Although the tests may be carried out correctly, their applicability is limited in the current form of the manuscript. I therefor asked the authors to adjust the manuscript more towards clinical use.

The authors chose to pool studies of different study desings in one meta-analyses. I answerd "partly" on question one, since I am not convinced that this is appropriate for the current study.

6. PLOS authors have the option to publish the peer review history of their article (what does this mean?). If published, this will include your full peer review and any attached files.

Reviewer #1: No

Reviewer #2: No

---

## [Author Response · Author response to Decision Letter 0]

17 Nov 2019

Response to reviewer # 1

1- Title: according to the results please add in the title “poor”

Response to reviewer: Thanks for your suggestion. We revised the title as recommended “Prediction of poor outcome after hypoxic-ischemic brain injury by diffusion-weighted imaging: A systematic review and meta-analysis”

2- Abstracts: “explicit” please find a synonymous. 

Response to reviewer: Thanks for your comment. We replaced “explicit” by “clear and generally accepted” in the revised abstract. 

3- Please use prognostic instead of diagnostic 

 Response to reviewer: Thanks for your comment. We used “prognostic” instead of “diagnostic” in the revised manuscript.

4- Introduction:

A- pg 3:

“disease modifying agents” this definition is not appropriate for HIE,

 Response to reviewer: Thanks for your comment. We deleted this phrase in the revised manuscript (page 3, lines20-21).

B- “normothermia phase which render the prognostic predictors less reliable[7].” This is not true, please have a loo to recent Bibliography, such as Scarpino et al., 2018, Resuscitation 

Response to reviewer: Thanks for your comment. We agree, low doses of sedatives and therapeutic hypothermia (TTM) had limited effects on cortical SEP components and/or the EEG (Scarpino, Lanzo et al. 2018). However, according to European Resuscitation Council and European Society of Intensive Care Medicine Guidelines for Post-resuscitation Care 2015, both TTM itself and sedatives or neuromuscular blocking drugs used to maintain it may potentially interfere with prognostication indices, especially those based on clinical examination (Nolan JP et al, 2015, Resuscitation). We rewrote this statement to make it clearer in the revised version of our manuscript (page 3, lines20-21). 

C- pg 4: developing a more accurate assessment of acute-stage HIBI patients is urgently needed.: I do not agree with these, because performing MRI is not possible usually in the first 24 hours.

 Response to reviewer: Thanks for your comment. Obtaining a brain MRI in critically-ill patients with potential cardiac instability may be challenging during the acute-stage (Wijdicks, Campeau et al. 2001). We agree that using the word “acute”may be not appropriate. Therefore, we used the term “early-stage” instead of “acute-stage”. However, we believe that the development of advanced equipment will enable the use of MRI examination at earlier HIBI stages. 

D- Nevertheless, CT and conventional MRI frequently underestimate the degree of brain injury in acute HIBI [12: being a systematic review I think not appropriate referring to a reference of 1999.

Response to reviewer: Thanks for your suggestion. We updated the relevant literature.

5- Methods

A- Pg 4:

In this study, we performed a comprehensive literature research in PubMed, EMBASE, and the Cochrane Library databases for DWI from January 1995 to December 2018.: this point seems to be very illogical for two mean reasons 1) TTM have been introduced about 2002 2) DWI is not a technique introduced so early. Actually only one reference is dated 2001.

 Response to reviewer: Thanks for your comment. The clinical application of DWI for ischemic brain injury began in the 1990s (Fisher, Prichard et al. 1995, Schabitz and Fisher 1995). In our preliminary search, we found that the first study reporting the application of DWI in global cerebral ischemia was published in 1999 (Arbelaez, Castillo et al. 1999) Therefore, for the comprehensiveness of our search, we set the beginning time as 1995 and this did not influence our results. 

Regarding TTM treatment, subjects included in our analysis were not limited to patients receiving TTM treatment. In fact, all eligible subjects with hypoxic–ischemic brain injury were included in this study regardless they received TTM treatment or no. I hope this explanation clarifies our point of view.

B- Pg 5: The outcome was classified into poor and good according to the CPC scores (3-4 or 4-5 versus1-2 or 1-3, respectively). Alternatively, we classified the outcomes into CPC 4–5 versus 1–3, if the CPC thresholds were not defined.

Please clarify this point, the authors extrapolate the data from the table presented by authors according to the cut off they used. I do not agree to attribute a priori a cut off if this is not specified and the studies not reporting clear indication should be not considered in the analysis.

 Response to reviewer: Thanks for your comment. Among the enrolled studies, the cutoff value was not defined in only one study (Wijdicks, Campeau et al. 2001). In that study, the outcome for each patient was recorded for analysis. On the other hand, cut off values were defined in all the remaining studies. Therefore, we were able to define the prognostic results for each patient and thus our definition of CPC 4–5 as poor outcome was based on the results reported by the original authors.

 In order to obtain more reliable results, we set more stringent inclusion criteria for this research by excluding studies that involved less than 10 patients. Therefore, we excluded three small studies and thus the above statement was deleted. (p5, line 21)

C- Table 2: I’m very surprised to find in the table studies in which idex test has specificity and sensitivity of 100%. He authors reviewed the quality of the study and how can include these studies in their computation?

 Response to reviewer: Thanks for your comment. According to our inclusion and exclusion criteria, we initially enrolled 29 observational cohort studies for our final analysis, which included 13 prospective studies and 16 retrospective studies. All 16 retrospective studies were cohort studies and the data of 6 out of those 16 studies were prospectively collected and retrospectively analyzed. 

To update our data, we re-searched the databases (PubMed, EMBASE, and the Cochrane Library databases) from January 1995 to September 2019, and set more stringent inclusion criteria for our analysis (i.e., studies with less than 10 patients were excluded). Accordingly, we found two additional studies (Velly, Perlbarg et al. 2018, Oren, Chang et al. 2019) and removed three original studies from the literature research due to the small number of cases (Wijdicks, Campeau et al. 2001, Heradstveit, Larsson et al. 2011, Choi, Youn et al. 2012). Next, we analyzed the diagnostic results (i.e. the true-positive, false-positive, false-negative, and true-negative results) for calculating their sensitivity and specificity. Finally, we observed that the main conclusion was still the same with our previous analysis. This further confirms the consistency of our results. The updated analysis has been included in the revised manuscript. 

7- DISCUSSION

A- PG 17

“Given this finding, the semi-quantitative method was used by only 2 studies”

Please rephrase this sentence, this is a systematic review, not an original paper reporting data about a sample of patients.

 Response to reviewer: We updated the data and revised this sentence (page 17 lines 13-20 of the revised manuscript).

B- PG 18

“Further, during brain damage, Wu et al., demonstrated an initial ADC reduction in the striatum and thalamus, followed by the cortex and the subcortical white matter. This DWI pattern could be an indication of ongoing tissue damage due to secondary apoptotic processes, and thus various brain structures can respond differently to ischemic injury [22].”

This sentence is not useful, in this kind of paper the authors should report consideration about cumulative data, is not a narrative report.

Response to reviewer: Thanks for your suggestion, we deleted this sentence and rewritten the relevant paragraph (page 16, lines 1-4 of the revised manuscript).

C- Pg 18-19

“However, our pooled analysis did not demonstrate that sensitivity and specificity of DWI were time dependent. The meta-regression analysis showed that the elapsed time between HIBI and brain MRI examination was not a factor in heterogeneity Interestingly, their diagnostic accuracy was not less stringent than other studies acquiring the DWI data during the ideal time window. This suggests that the time of the MRI is not an important factor in determining diagnostic accuracy.

This is an example of what I have underlined in the first part of my comments; this results is true if we evaluated the value of specificity, in this case 100%, but this finding would be right only if in all the paper the authors had used the same parameters to reach the best predictive power at every window. In fact the studies reported by the authors [20, 38, 44, 48, 51, 52] all used different measure od MRI , so the message that time dependence of MRI is not a factor is not true. MRI is a time dependent test and for every time frame require different measure to reach the best predicitve power.

 Response to reviewer: Thanks for your important comment. We re-searched the databases (PubMed, EMBASE, and the Cochrane Library databases) from January 1995 to September 2019. We included 2 new studies (Velly, Perlbarg et al. 2018, Oren, Chang et al. 2019), and excluded three original because of the small number of cases (Wijdicks, Campeau et al. 2001, Heradstveit, Larsson et al. 2011, Choi, Youn et al. 2012). Our subgroup analysis showed that diagnostic accuracy during 6 days after HIBI was higher than that after 6 days, although our meta-regression analysis showed that the elapsed time between HIBI and brain MRI examination was not a factor in heterogeneity (Table 1). Interestingly, the diagnostic accuracy during the first 24 hours after HIBI was not less stringent than the other studies acquiring the DWI data during days 2 to 6. This does not contradict the previous conclusion that DWI signal abnormalities were time dependent, because those studies used different diagnostic strategy and criteria (Hirsch, Mlynash et al. 2015, Park, Lee et al. 2015). Although DWI signals were time-dependent but, different diagnostic strategies could be used to improve the accuracy of diagnosis. It is worth mentioning that the diagnostic accuracy of DWI will significantly decrease if the examination period exceeds 7 days. We revised the manuscript accordingly (page16, lines 4-15). 

 Table 1. Subgroup analysis among the different study subsets.

Study characteristics No of subsets Pooled sensitivity(95 % CI) Pooled specificity(95 % CI) P

Total 98 0.613(0.599-0.628) 0.958(0.947-0.967) 

Time of MRI examination 0.1426

~1d 9 0.767(0.694-0.829) 1.000(0.960-1.000) 

2~6d 85 0.627(0.611-0.642) 0.953(0.941-0.963) 

>6d 4 0.425(0.376-0.475) 0.991(0.949-1.000) 

D- PG 19

“Therefore, DWI examination could be carried out in a wider timeframe than other prognostic strategies like clinical examination, myoclonus and status myoclonus, electroencephalogram, or biomarkers”

Again this is not completely true, EEG and SEP can be performed in any time windows, and offer also the advantage to be more available in every clinical setting, can be recorded bed-side and repeated more time.

 Response to reviewer: Thanks a lot. We agree, and we rewritten the paragraph accordingly (page 16, lines 9-21; page 17, lines 1-12 )

8- CONCLUSION

They need to be completely rephrased according to the revised version of the manuscript according to the point raised.

 Response to reviewer: Thanks for your valuable suggestions. We rewritten this paragraph accordingly in the revised version (pages 19, lines 9-13).

Response to reviewer #2

1- In my opinion, the authors have performed many statistical tests, but with limited adjustments to the clinical, technical and pathofysiological background of the DWI analyses. Although the tests may be carried out correctly, their applicability is limited in the current form of the manuscript. I therefor asked the authors to adjust the manuscript more towards clinical use.

Response to the reviewer：Thanks for suggestion. In order to make this manuscript more comprehensive and reliable for clinicians, we re-searched the databases (PubMed, EMBASE, and the Cochrane Library databases) from January 1995 to September 2019, and set more stringent inclusion criteria for our analysis. Studies that involved less than 10 patients were excluded. Accordingly, we included two additional studies (Velly, Perlbarg et al. 2018, Oren, Chang et al. 2019) and removed three original studies from the literature research due to the small number of cases (Wijdicks, Campeau et al. 2001, Heradstveit, Larsson et al. 2011, Choi, Youn et al. 2012). We re-analyzed the included data, then added new statements and revised the discussion section to make the manuscript more informative for clinicians. 

In 2015 , the European Society of Intensive Care Medicine and European Resuscitation Council (Nolan JP, et al，Resuscitation, 2015.) highlighted the limitations of studies using MRI after HIBI. The lack of homogeneity in radiological definitions of imaging findings and neuroimaging caused this drawback. Our review focused on DWI research and it included recently published studies to help clinicians make informed decisions based on the summary of studies performed in the last 25 years. Further, we added meta-regression and subgroup analysis to explain the limitations of the enrolled studies thus, providing a more accurate theoretical basis for the clinical analysis and judgment.

2- The authors chose to pool studies of different study desings in one meta-analyses. I answerd "partly" on question one, since I am not convinced that this is appropriate for the current study.

Response to reviewer：Thanks for suggestion again. According to our inclusion and exclusion criteria, we initially enrolled 29 observational cohort studies for our final analysis which included 13 prospective studies and 16 retrospective studies. All 16 retrospective studies were cohort studies, and data of 6/16 studies were prospectively collected and retrospectively analyzed. Our preliminary analysis suggested the absence of significant difference in the diagnostic accuracy between the prospective group and retrospective group. Therefore, we included all these observational cohort studies in our analysis. 

To update our data, we also re-searched the databases and set more stringent inclusion criteria for our analysis. In addition, we also carefully checked and explored the study heterogeneity (the variability across studies). We evaluated the threshold effect of included studies at first and then we performed a meta-regression analysis and subgroup analysis to investigate factors that could possibly lead to heterogeneity and explored their possible impact on diagnostic accuracy. This is also critical for our clinical analysis and further research. (page 8, lines 4-5; p16-19).

References 

Arbelaez, A., M. Castillo and S. K. Mukherji (1999). "Diffusion-weighted MR imaging of global cerebral anoxia." American Journal of Neuroradiology 20(6): 999-1007.

Choi, S. P., C. S. Youn, K. N. Park, J. H. Wee, J. H. Park, S. H. Oh, S. H. Kim and J. Y. Kim (2012). "Therapeutic hypothermia in adult cardiac arrest because of drowning." Acta Anaesthesiol Scand 56(1): 116-123.

Fisher, M., J. W. Prichard and S. Warach (1995). "New magnetic resonance techniques for acute ischemic stroke." Jama 274(11): 908-911.

Heradstveit, B. E., E. M. Larsson, H. Skeidsvoll, S. M. Hammersborg, T. Wentzel-Larsen, A. B. Guttormsen and J. K. Heltne (2011). "Repeated magnetic resonance imaging and cerebral performance after cardiac arrest--a pilot study." Resuscitation 82(5): 549-555.

Hirsch, K. G., M. Mlynash, S. Jansen, S. Persoon, I. Eyngorn, M. V. Krasnokutsky, C. A. Wijman and N. J. Fischbein (2015). "Prognostic value of a qualitative brain MRI scoring system after cardiac arrest." J Neuroimaging 25(3): 430-437.

Oren, N. C., E. Chang, C. W. Y. Yang and S. K. Lee (2019). "Brain Diffusion Imaging Findings May Predict Clinical Outcome after Cardiac Arrest." Journal of Neuroimaging 29(4): 540-547.

Park, J. S., S. W. Lee, H. Kim, J. H. Min, J. H. Kang, K. S. Yi, K. H. Park and B. K. Lee (2015). "Efficacy of diffusion-weighted magnetic resonance imaging performed before therapeutic hypothermia in predicting clinical outcome in comatose cardiopulmonary arrest survivors." Resuscitation 88: 132-137.

Scarpino, M., G. Lanzo, F. Lolli, R. Carrai, M. Moretti, M. Spalletti, M. Cozzolino, A. Peris, A. Amantini and A. Grippo (2018). "Neurophysiological and neuroradiological multimodal approach for early poor outcome prediction after cardiac arrest." Resuscitation 129: 114-120.

Schabitz, W. R. and M. Fisher (1995). "Diffusion weighted imaging for acute cerebral infarction." Neurol Res 17(4): 270-274.

Velly, L., V. Perlbarg, T. Boulier, N. Adam, S. Delphine, C. E. Luyt, V. Battisti, G. Torkomian, C. Arbelot, R. Chabanne, B. Jean, C. Di Perri, S. Laureys, G. Citerio, A. Vargiolu, B. Rohaut, N. Bruder, N. Girard, S. Silva, V. Cottenceau, T. Tourdias, O. Coulon, B. Riou, L. Naccache, R. Gupta, H. Benali, D. Galanaud, L. Puybasset, J. M. Constantin, J. Chastre, J. Amour, C. Vezinet, J. J. Rouby, M. Raux, O. Langeron, V. Degos, F. Bolgert, N. Weiss, T. Similowski, A. Demoule, A. Duguet, E. Tollard, B. Veber, J. A. Lotterie, P. Sanchez-Pena, M. Génestal and M. Patassini (2018). "Use of brain diffusion tensor imaging for the prediction of long-term neurological outcomes in patients after cardiac arrest: a multicentre, international, prospective, observational, cohort study." The Lancet Neurology 17(4): 317-326.

Wijdicks, E. F., N. G. Campeau and G. M. Miller (2001). "MR imaging in comatose survivors of cardiac resuscitation." AJNR Am J Neuroradiol 22(8): 1561-1565.

Thanks again for your meticulous reviews. All new amendments are written in red font color in the revised version of our manuscript. I hope the revised version is now acceptable for publication.

---

## [Editor Report · Decision Letter 1]

25 Nov 2019

Prediction of poor outcome after hypoxic-ischemic brain injury by diffusion-weighted imaging: A systematic review and meta-analysis

PONE-D-19-17572R1

Dear Dr. Luo,

We are pleased to inform you that your manuscript has been judged scientifically suitable for publication and will be formally accepted for publication once it complies with all outstanding technical requirements.

With kind regards,

Chiara Lazzeri

Academic Editor

PLOS ONE
---

## [Editor Report · Acceptance letter]

10 Dec 2019

PONE-D-19-17572R1 

Prediction of poor outcome after hypoxic-ischemic brain injury by diffusion-weighted imaging: A systematic review and meta-analysis 

Dear Dr. Luo:

I am pleased to inform you that your manuscript has been deemed suitable for publication in PLOS ONE. Congratulations! Your manuscript is now with our production department. 

With kind regards,

on behalf of

Dr. Chiara Lazzeri 

Academic Editor

PLOS ONE